# Evaluation of the Antibacterial and Prebiotic Potential of *Ascophyllum nodosum* and Its Extracts Using Selected Bacterial Members of the Pig Gastrointestinal Microbiota

**DOI:** 10.3390/md20010041

**Published:** 2021-12-30

**Authors:** Brigkita Venardou, John V. O’Doherty, Marco Garcia-Vaquero, Claire Kiely, Gaurav Rajauria, Mary J. McDonnell, Marion T. Ryan, Torres Sweeney

**Affiliations:** 1School of Veterinary Medicine, University College Dublin, Belfield, D04 V1W8 Dublin 4, Ireland; brigkita.venardou@ucdconnect.ie (B.V.); marion.ryan@ucd.ie (M.T.R.); 2School of Agriculture and Food Science, University College Dublin, Belfield, D04 V1W8 Dublin 4, Ireland; john.vodoherty@ucd.ie (J.V.O.); marco.garciavaquero@ucd.ie (M.G.-V.); ckiel08@outlook.com (C.K.); gaurav.rajauria@ucd.ie (G.R.); marymcdonnell06@gmail.com (M.J.M.)

**Keywords:** *Ascophyllum nodosum*, seaweed polysaccharides, fucoidan, antibacterial, prebiotic, *Bifidobacterium*, *Enterobacteriaceae*

## Abstract

*Ascophyllum nodosum* and its extracts are promising antibacterial and prebiotic dietary supplements for pigs. The objectives of this study were to evaluate the effects of the increasing concentrations of: (1) two whole biomass samples of *A. nodosum* with different harvest seasons, February (ANWB-F) and November (ANWB-N), in a weaned pig faecal batch fermentation assay, and (2) *A. nodosum* extracts produced using four different extraction conditions of a hydrothermal-assisted extraction methodology (ANE1–4) and conventional extraction methods with water (ANWE) and ethanol (ANEE) as solvent in individual pure culture growth assays using a panel of beneficial and pathogenic bacterial strains. In the batch fermentation assay, ANWB-F reduced *Bifidobacterium* spp. counts (*p* < 0.05) while ANWB-N increased total bacterial counts and reduced *Bifidobacterium* spp. and *Enterobacteriaceae* counts (*p* < 0.05). Of the ANE1–4, produced from ANWB-F, ANWE and ANEE that were evaluated in the pure culture growth assays, the most interesting extracts were the ANE1 that reduced *Salmonella* Typhimurium, enterotoxigenic *Escherichia coli* and *B. thermophilum* counts and the ANE4 that stimulated *B. thermophilum* growth (*p* < 0.05). In conclusion, the extraction method and conditions influenced the bioactivities of the *A. nodosum* extracts with ANE1 and ANE4 exhibiting distinct antibacterial and prebiotic properties in vitro, respectively, that merit further exploration.

## 1. Introduction

The gastrointestinal microbiota performs a wide range of biologically important functions for the host, such as increasing nutrient availability, immunomodulation and inhibition of pathogen colonisation [1,2,3]. The *Lactobacillus* and *Bifidobacterium* genera are among the best-characterised beneficial members in terms of their contributions to these functions in a range of mammals, including humans and pigs [4,5,6,7,8,9]. Another important constituent is the *Enterobacteriaceae* family, as it harbours many pathogens such as pathogenic *Escherichia coli* strains and *Salmonella enterica* subsp. *enterica* serotypes (e.g., Typhimurium, Enteritidis) that can lead to enteric and systemic disease in humans and animals [10,11]. Disturbances in the composition of the gastrointestinal microbiota can result in dysbiosis, a transient or permanent pathological state, characterised by the loss of beneficial bacterial groups, the overgrowth of opportunistic pathogens and/or loss of overall microbial diversity [12]. An increase of the *Enterobacteriaceae* is among the common compositional changes that characterise dysbiotic microbiota in pigs and humans [13,14,15]. 

Dietary interventions can promote a more beneficial composition of the gastrointestinal microbiota and reduce the incidence of dysbiosis [12]. Brown macroalgae or seaweeds (class *Phaeophyceae*) are rich in non-digestible polysaccharides (alginates, laminarins, fucoidans, mannitol), minerals, polyphenols, and essential amino acids, all associated with a wide range of biological activities [16,17]. Of interest to this study are the prebiotic [18,19] and antibacterial [20,21] potentials of the brown seaweed extracts and, particularly, the non-digestible polysaccharides fractions. However, the quantity, chemical structure and bioactivity of seaweed polysaccharides is variable and dependent on numerous factors, including: species, season of harvest, geographic location, environmental conditions and the extraction methodologies and conditions used [20,21,22]. In conventional extraction methodologies, the typically used solvents are water for the extraction of soluble sulphated polysaccharides such as fucoidans [23], and ethanol for extracts rich in polyphenols [24]. Despite its convenience, solvent extraction is time-consuming and inefficient in terms of the quantity and quality of bioactives obtained, including polysaccharides [25]. Therefore, novel extraction methodologies aim to optimise extraction efficiency while also reducing production time and cost. This can be facilitated by the use of the response surface methodology, a multivariate statistic technique, with which one or more extraction parameters (pH, time, temperature, pressure, and solvent-to-seaweed ratio) are selected to be improved to achieve the highest concentration and/or preferred chemical structure of a chosen seaweed component [22]. For example, the conditions of a hydrothermal-assisted extraction (HAE) method were recently optimised using the response surface methodology to obtain the best concentration of fucoidan and/or laminarin and/or antioxidant activity from brown seaweeds [26].

In the current study, the antibacterial and prebiotic potential of the brown seaweed *Ascophyllum nodosum* was investigated for two reasons. Firstly, this seaweed species has a high polysaccharide content, characterised by monthly fluctuation ranging from ≈50% in February to ≈70% in November [27]. Furthermore, dietary supplementation of whole *A. nodosum* biomass or its extracts has consistently been associated with reductions in *E. coli* numbers and alterations in the composition of the gastrointestinal microbiota in both in vitro and in vivo models [28,29,30,31,32,33]. The use of in vitro assays such as the batch fermentation and pure culture growth assays are considered practical screening tools for evaluating the effect of candidate compounds on selected bacterial populations at the family, genus and species levels [34,35]. Thus, the objectives of this study were to investigate: (1) whether the harvest season of the whole biomass samples of *A. nodosum* will have an effect on selected intestinal bacterial populations in a batch fermentation assay; and (2) how the conventional extraction methods and the different conditions of a novel HAE methodology alter the antibacterial and prebiotic activities of *A. nodosum* extracts in a panel of pure culture growth assays.

## 2. Results

### 2.1. Proximate Composition of Whole A. nodosum Biomass Samples and Composition of A. nodosum Extracts

The proximate compositions of the two dried whole seaweed biomass samples are presented in Table 1 as previously published by Garcia-Vaquero, et al. [36].

The laminarin and fucoidan content of ANE1–4 and the total soluble sugars and fucoidan content of ANWE and ANEE are presented in Table 2.

### 2.2. Effects of Fructooligosaccharides (FOS), ANWB-F and ANWB-N on the Selected Faecal Bacterial Populations

The effects of FOS (prebiotic used as a control) and whole biomass samples of *A. nodosum* collected in February (ANWB-F) and November (ANWB-N) on selected faecal bacterial populations are described below and presented in Table 3.

FOS: There was an increase in total bacterial (*p* < 0.05) and *Bifidobacterium* spp. (*p* = 0.053) counts at 2.5 and 5 mg/mL at 10 h compared to the control (*p* ≤ 0.05), but not at 24 h (*p* > 0.05). FOS had no effect on the counts of *Lactobacillus* spp. and *Enterobacteriaceae* at all tested concentrations and time points (*p* > 0.05).

ANWB-F: There was a decrease in *Bifidobacterium* spp. counts at 2.5 and 5 mg/mL at 10 (*p* = 0.057) and 24 h (*p* < 0.05) compared to the control. It is worth noting that the inclusion of 5 mg/mL ANWB-F reduced *Bifidobacterium* spp. counts below the limit of detection of the QPCR (≈1.5 log transformed gene copy number/ 3 μL plasmid). ANWB-F had no effect on the counts of total bacteria, *Lactobacillus* spp. and *Enterobacteriaceae* at all tested concentrations and time points (*p* > 0.05).

ANWB-N: There was a decrease in *Bifidobacterium* spp. counts at 2.5 and 5 mg/mL at 10 h (*p* < 0.05) and at all ANWB-N concentrations at 24 h (*p* < 0.05) compared to the control. Furthermore, *Bifidobacterium* spp. counts dropped below the limit of detection of the QPCR (≈1.5 log transformed gene copy number/3 μL plasmid) at the concentrations of 2.5 and 5 mg/mL ANWB-N. There was an increase in total bacterial counts at all ANWB-N concentrations at 10 (*p* = 0.055) and 24 h (*p* < 0.05) compared to the control. There was no effect of any ANWB-N concentration on *Enterobacteriaceae* counts at 10 h (*p* > 0.05); however, a variable response to the different ANWB-N concentrations was observed at 24 h, with the reduction in *Enterobacteriaceae* counts at 5 mg/mL compared to the control being the most pronounced (*p* < 0.05). ANWB-N had no effect on the counts of *Lactobacillus* spp. at all tested concentrations and time points (*p* > 0.05), despite the observed increase in its counts at 1 and 2.5 mg/mL at 24 h compared to the 5 mg/mL (*p* < 0.05).

### 2.3. Antibacterial and Prebiotic Properties of A. nodosum Extracts in Pure Bacterial Cultures

The antibacterial and prebiotic activities of the *A. nodosum* extracts (ANWE, ANEE and ANE1–4) were evaluated in pure culture growth assays using a panel of selected beneficial (*L. plantarum*, *L. reuteri* and *B. thermophilum*) and pathogenic (ETEC and *S.* Typhimurium) bacterial strains and are presented in Table 4. The antibacterial activity against the ETEC strain was only evaluated for the ANE1–4 extracts due to the strong anti-*S.* Typhimurium activity of ANE1.

#### 2.3.1. Conventional Extraction Methods

ANWE: There was a quadratic effect of the increasing ANWE concentrations on the counts of *L. plantarum*, *L. reuteri* and *B. thermophilum* (*p* < 0.05). The concentrations of 0.5 and 2 mg/mL ANWE were associated with the greatest reduction in *L. plantarum* and *L. reuteri* counts, respectively. *B. thermophilum* reached its highest counts at 1 mg/mL. There was a linear increase in *S.* Typhimurium counts in response to the increasing ANWE concentrations (*p* < 0.05).

ANEE: There was a linear increase in *L. plantarum* counts and a linear decrease in *L. reuteri* and *B. thermophilum* counts in response to the increasing ANEE concentrations (*p* < 0.05). ANEE had no effect on the counts of *S.* Typhimurium at all concentrations tested (*p* > 0.05).

#### 2.3.2. HAE Methodology

ANE1: There was a linear increase in *L. reuteri* counts in response to the increasing ANE1 concentrations (*p* < 0.05). There was a linear decrease in *B. thermophilum* counts in response to the increasing ANE1 concentrations (*p* < 0.05). There was a beneficial quadratic effect of the increasing ANE1 concentrations on ETEC and *S.* Typhimurium, with the lowest counts for both bacterial strains observed at 2 mg/mL (*p* < 0.05). ANE1 had no effect on the counts of *L. plantarum* at all concentrations tested (*p* > 0.05). 

ANE2: There was a quadratic effect of the increasing concentrations of ANE2 on *B. thermophilum* and ETEC counts (*p* < 0.05). *B. thermophilum* counts were reduced at all concentrations above 1 mg/mL (*p* < 0.05). In the case of ETEC, its counts were initially increased at the lower ANE2 concentrations before reducing at 2 mg/mL (*p* < 0.05). There was a linear decrease in *S.* Typhimurium counts (*p* < 0.05). ANE2 had no effect on the counts of *L. plantarum* and *L. reuteri* at all concentrations tested (*p* > 0.05).

ANE3: ANE3 had no effect on the counts of all bacterial strains tested (*p* > 0.05).

ANE4: There was a linear increase in *L. reuteri* counts in response to the increasing ANE4 concentrations (*p* < 0.05). There was a beneficial quadratic effect of the different ANE4 concentrations observed on *B. thermophilum,* with maximum counts observed at 1 mg/mL (*p* < 0.05). ANE4 had no effect on the counts of *L. plantarum*, ETEC and *S.* Typhimurium at all concentrations tested (*p* > 0.05).

## 3. Discussion

In this study, the influence of harvest season on the effects of the whole biomass samples of *A. nodosum* on selected intestinal bacterial populations was evaluated in a batch fermentation assay inoculated with pig faeces. ANWB-F was solely associated with reduced *Bifidobacterium* spp. counts, whereas ANWB-N increased total bacterial counts and reduced *Bifidobacterium* spp. and *Enterobacteriaceae* counts. These observations suggest that seasonal variation in the bioactivity of *A. nodosum* may exist. To evaluate the effects of the different extraction methods and conditions on the antibacterial and prebiotic potential of the *A. nodosum* extracts, pure culture growth assays using a panel of selected beneficial and pathogenic bacterial strains were employed. The *A. nodosum* extracts tested were the ANE1–4 produced from the least antibacterial ANWB-F using the four different conditions of the novel HAE methodology, as well as the ANWE and ANEE produced using conventional extraction methodologies. The two most interesting of these extracts were ANE1 and ANE4. ANE1 displayed strong antibacterial activity against *S.* Typhimurium and ETEC, but also reduced *B. thermophilum* counts. ANE4 had a prebiotic effect as it stimulated the growth of *B. thermophilum*. 

Whole biomass samples of *A. nodosum* collected in February and November (ANWB-F and ANWB-N) were evaluated in the batch fermentation assay to ascertain their effects on total bacteria, *Bifidobacterium* spp., *Enterobacteriaceae* and *Lactobacillus* spp. in the faecal microbiota of weaned pigs. FOS was also included as a prebiotic control. As expected, FOS increased total bacterial and *Bifidobacterium* spp. counts in agreement with another batch fermentation study using pig faeces [37] and weaned pig studies [38,39]. Both ANWB-F and ANWB-N reduced *Bifidobacterium* spp. counts in a clear concentration-dependent manner, with the latter having a stronger effect. A linear reduction in *Bifidobacterium* spp. counts and the relative abundance of the Actinobacteria phylum, which includes the *Bifidobacterium* genus in response to the increasing dietary levels of *A. nodosum,* air-dried whole biomass sample and crude extract, has previously been observed in pigs and rams [28,32]. ANWB-N additionally increased total bacterial counts and reduced *Enterobacteriaceae* counts. The inhibitory effects of *A. nodosum*, whole biomass samples and crude extracts on *Enterobacteriaceae* has also been supported by in vitro and in vivo studies [28,29,30,32]. The findings above indicate that harvest season influenced the effects of *A. nodosum* on the faecal microbiota. Therefore, this should be further explored in future studies on the seasonal variation of *A. nodosum* bioactivity.

The main polysaccharide constituent of the *A. nodosum* samples was fucoidan without any significant variation in concentration between February and November, in agreement with Fletcher, et al. [40]. Fucoidans are a group of heterogenous sulphated polysaccharides in brown seaweeds, with a backbone of α-linked-L-fucose residues (α-1,3 or alternating α-1,3/α-1,4) and varying degrees of branching and sulphate and other monosaccharide contents [41]. While there was no difference in fucoidan content between ANWB-F and ANWB-N samples, it remains to be determined if there were structural differences (monosaccharide composition, sulfation level) in the fucoidan due to seasonality [40,42,43,44] that could explain the variability in the effects on *Bifidobacterium* spp. and *Enterobacteriaceae* counts in the batch fermentation assay. The previously reported antibacterial activity of this polysaccharide against a range of Gram-negative and Gram-positive bacterial species supports the assumption that fucoidan was probably the bioactive influencing the observed reduction in *Bifidobacterium* spp. and *Enterobacteriaceae* counts [45]. In the case of *Bifidobacterium* spp., the observed reduction in their counts could be the result of a combination of direct antibacterial activity and inhibition of the bacterial carbohydrate-degrading enzymes due to fucoidan [46] and the phenolic fraction [47,48]. However, the bioactive components of the whole biomass samples of *A. nodosum* that were responsible for the observed effects on the faecal bacterial populations were not identified in the current study.

In the second part of the study, the ANWB-F, possessing the least antibacterial activity, was selected to produce ANE1–4 to determine whether the different conditions of the novel HAE methodology would result in extracts with improved bioactivity. ANWE and ANEE, both produced using conventional extraction methods, were also included. The influence of the extraction methodology on the antibacterial and prebiotic activity of *A. nodosum* was investigated in a panel of pure culture growth assays. Two representative pathogens were selected from the *Enterobacteriaceae* family; *S.* Typhimurium, a major foodborne pathogen, with pork products as a common source of human infection [49], and ETEC which is involved in the development of post-weaning diarrhoea in commercially weaned pigs [50]. The effects of *A. nodosum* extracts on three beneficial bacterial strains (*L. plantarum*, *L. reuteri* and *B. thermophilum*) was also evaluated. *L. plantarum*, *L. reuteri* and *B. thermophilum* are indigenous members of the gastrointestinal microbiota in humans and pigs with important contributions to colonisation resistance against intestinal pathogens, immunomodulation and maintenance of intestinal integrity [51,52,53,54,55,56,57,58,59,60]. From the conventionally produced *A. nodosum* extracts, ANWE was associated with an increase in *B. thermophilum* and *S.* Typhimurium counts in addition to a reduction in both *Lactobacillus* spp. strains, whereas ANEE increased *L. plantarum* but reduced the other two beneficial strains. The next set of extracts evaluated, ANE1–4, were produced using the different conditions of the HAE methodology. ANE1 was highly potent in reducing the counts of both pathogens, while ANE2 exhibited some antibacterial activity against *S.* Typhimurium and ETEC. ANE1 and ANE2 also maintained the antibacterial activity of the parent ANWB-F against *Bifidobacterium* spp., as they both reduced *B. thermophilum*. However, ANE1 slightly increased the counts of *L. reuteri*. Interestingly, ANE4 displayed a strong bifidogenic activity with an additional stimulation of *L. reuteri* growth but had no inhibitory activity against *S.* Typhimurium and ETEC. ANE3 was the least effective *A. nodosum* extract with no effect on the growth of any of the tested bacterial strains. Taking all of the above together, the two most potent extracts produced using the HAE methodology were the ANE1 (antibacterial) and ANE4 (prebiotic), both of which are promising candidates for dietary modulation of the gastrointestinal microbiota. Contrarily, the variable effects of ANWE and ANEE (obtained from the conventional extraction methods) indicate that these extracts are unsuitable for such applications.

While the antibacterial and bifidogenic components of ANE1 and ANE4, respectively, were not been determined in this study, we hypothesise that these activities were probably associated with fucoidan which represented approximately a third of the ANE1–4. The variation in bioactivity between extracts was most likely a result of changes in the chemical structure of fucoidan due to the different extraction conditions (Table 5). Alterations in the chemical composition and sulphate content, as well as lower molecular weight due to partial hydrolysis have been reported in relation to extracted fucoidan from *A. nodosum* in response to the increasing extraction temperature and the application of HCl as solvent [61,62]. The assumption that the variation in antibacterial activity is associated with the structural changes of fucoidan is further supported by the increased antibacterial activity of fucoidan from *Laminaria* spp., *Sargassum* spp. and *Undaria* spp. after depolymerization [45,63,64,65]. Depolymerised fucoidan is considered to exert its antibacterial activity by interfering with the integrity and permeability of the bacterial cell membrane, resulting in cell death [45,63] and by nutrient trapping resulting in reduced nutrient availability [64]. Furthermore, the direct antibacterial activity of fucoidan was found to be stronger against Gram-negative than Gram-positive bacteria [63], hence the higher reduction in *S.* Typhimurium and ETEC compared to *B. thermophilum* in response to ANE1. Regarding the bifidogenic activity of ANE4, it has previously been reported that the bifidobacterial genome encodes for various glycoside hydrolase families including α-L-fucosidases, thus enabling *Bifidobacterium* spp. to utilise fucoidan as a substrate [7,19]. Furthermore, low molecular weight fucoidans from *Laminaria* spp. and *Sargassum* spp. were associated with improved *Bifidobacterium* spp. counts in a batch fermentation with human faeces [66] and in a pure culture growth assay [67]. All the above support our assumption that it is the fucoidan in the ANE4 that is principally responsible for the increase in *B. thermophilum* counts in the pure culture assay. The ability of some members of the *Lactobacillus* genus to partially utilise fucoidan from *A. nodosum* probably explains the minor increase in *L. reuteri* counts observed in this study [68]. Based on the above, fucoidan was most likely the bioactive component in ANE1 and ANE4 associated with the antibacterial and prebiotic activities, with the variation between ANE1–4 attributed to structural alterations in this polysaccharide due to the different extraction conditions of HAE methodology. However, further compositional characterisation of ANE1 and ANE4 is required to confirm these assumptions.

A major drawback of conventional extraction methods is the inefficient extraction of seaweed polysaccharides [25]. This was evident in the ANWE and ANEE, both of which had a lower polysaccharide content compared to ANE1–4. The poorer performance of ANWE and ANEE could be a consequence of this low polysaccharide content as well as the presence of other polysaccharides apart from fucoidan. Compositional characterisation of ANWE and ANEE would have provided a better understanding of their effects on bacterial growth in this study; however, these extracts were not evaluated further due to their largely negative impact on the beneficial bacterial strains and small quantities. 

This initial screening study provided some information on the antibacterial and prebiotic potential of *A. nodosum* extracts. As this seaweed species is indigenous to many countries including Ireland, such extracts can be produced at an industrial scale as animal dietary supplements. However, further exploration is required on whether the extraction methodology used in this study can consistently produce bioactive extracts regardless of the variability in the composition of the whole seaweeds.

## 4. Materials and Methods

### 4.1. Whole Biomass Samples and Extracts of A. nodosum

A summary of preparation conditions for the whole biomass samples and extracts of *A. nodosum* that were utilised in this study is presented in Table 5, with further details as follows:

Whole seaweed biomass: *A. nodosum* was harvested in February (ANWB-F) and November (ANWB-N) once and from the same collection site by Quality Sea Veg Ltd. (Burtonport, Co., Donegal, Ireland). Samples were oven-dried, milled and stored at room temperature and their proximate composition (dry matter, ash, protein, crude lipids, total soluble sugars, total glucans, fucoidan and total phenolic content) was determined as described previously [36]. The dried whole biomass samples were tested in the fermentation assay. 

Conventional extraction methods: Extracts from *A. nodosum* were produced using a solvent solid-liquid extraction method using water (ANWE) and 80% ethanol: 20% water (ANEE) as described previously [69]. These extracts were freeze-dried and tested in the pure culture growth assays.

HAE methodology: Four extracts of ANWB-F were produced using a HAE methodology with optimised extraction conditions (temperature, incubation time and solvent (0.1 M HCl) to seaweed ratio), as described previously by Garcia-Vaquero, et al. [26]. Each extraction condition was optimised to give the best concentration of fucoidan (E1), laminarin (E2), antioxidant activity (E3) and all the above (E4). These extracts (ANE1–4) were freeze-dried and tested in the pure culture growth assays.

Polysaccharide content of *A. nodosum* extracts: The total soluble sugars of ANWE and ANEE were determined following the phenol-sulfuric acid assay as described previously [70]. The concentration of laminarin in ANE1–4 was determined with high-performance liquid chromatography (HPLC) (Varian *Prostar* HPLC system, Agilent Technologies, Santa Clara, CA, USA) using a calibration curve of commercial laminarin from *Laminaria digitata* (Sigma-Aldrich, St. Louis, MO, USA) as described by Rajauria, et al. [71]. The concentration of fucoidan in the ANE1-4, ANWE and ANEE was determined according to the method described by Usov et al. [72], with modifications as described by Garcia-Vaquero, et al. [26]. All samples were analysed on two independent occasions (2 biological replicates) with three readings each time.

### 4.2. Batch Fermentation Assay

The batch fermentation assay was carried out as described previously by Venardou et al. [73]. Briefly, faeces from 29 newly weaned crossbred pigs (Large White x Landrace) from a commercial farm (Perma pigs Limited, Co., Kildare, Ireland) were collected and pooled. The commercial farm had a *Salmonella* seroprevalence of 18%, with no other diseases important to pig health and production being reported. Weaned pigs were fed a cereal- and milk-based diet. The faecal inoculum (FI) was prepared one day prior to the batch fermentation assay by performing a 5-fold dilution of the pooled faeces on a weight basis (1:5 *w*/*v*) in pre-reduced phosphate-buffered saline (Sigma-Aldrich, St. Louis, MO, USA) and stored anaerobically at 4 °C. The FI was added to the fermentation medium at a 1:10 *v*/*v* ratio. ANWB-F and ANWB-N were diluted in a final volume of 21 mL FI/fermentation medium at 1, 2.5 and 5 mg/mL concentrations. FOS from chicory (F8052-50G) was also included at these concentrations due to its established prebiotic effects [74]. The batch fermentation was carried out in glass tubes (PYREX™ Disposable Round-Bottom Rimless Glass Tubes, Fisher Scientific, Co. Dublin, Ireland) with rubber stoppers (Saint-Gobain Rubber stopper grey blue 17/22 x H 25MM, VWR, Co. Dublin, Ireland). Anaerobic conditions were established and maintained by the addition of oxyrase (Sigma-Aldrich, St. Louis, MO, USA) and CO_2_ flushing. Control tubes containing only FI and fermentation medium were also included. All tubes were incubated at 39 °C for 24 h with gentle stirring (100 rpm). A volume of 5 mL fermentation broth was collected at 0, 10 and 24 h in duplicate tubes. The collected samples were centrifuged at 12,000× *g* for 5 min. The resultant pellets were stored in −20 °C until further analysis. All experiments were repeated on three independent occasions, hence three biological replicates.

### 4.3. Quantification of Bacterial Groups Using Quantitative Real-Time Polymerase Chain Reaction (QPCR)

*DNA extraction:* Microbial genomic DNA was extracted from the pellets using a QIAamp Fast DNA stool mini kit (Qiagen, West Sussex, UK) according to the manufacturer’s instructions. The DNA quantity and quality were evaluated using a Nanodrop spectrophotometer (Thermo Fisher Scientific, Waltham, MA, USA).

*Bacterial primers:* The primers targeting the 16S rRNA gene of total bacteria, *Lactobacillus* spp., *Bifidobacterium* spp. and *Enterobacteriaceae* were available in the literature and are provided in Table 6. Primers were designed using two tools, Primer3 (https://primer3.org/, accessed on 26 June 2018) for larger amplicons (>150 bp) and Primer Express™ (Applied Biosystems, Foster City, CA, USA) for smaller amplicons (<125 bp), and their specificity was verified using the Primer Basic Local Alignment Search Tool (Primer-BLAST), https://www.ncbi.nlm.nih.gov/tools/primer-blast/index.cgi, accessed on 26 June 2018.

*Plasmid-based standard curves and QPCR:* The preparation of specific plasmids for total bacteria, *Lactobacillus* spp., *Bifidobacterium* spp. and *Enterobacteriaceae* for the production of the respective standard curves and the QPCR for the quantification of the above-mentioned bacterial groups were performed as described in our previous work [73]. Briefly, bacterial genomic DNA from *Bifidobacterium thermophilum* (DSMZ 20210), *Lactobacillus plantarum (*DSMZ 20174) and *S.* Typhimurium PT12 was extracted from pure cultures using DNeasy^®^ Blood & Tissue kit (Qiagen, West Sussex, UK). The targeted genes were amplified using conventional PCR and incorporated into a vector using the TOPO™ TA Cloning™ Kit for Sequencing (Invitrogen, Thermo Fisher Scientific, Carlsbad, CA, USA). One Shot™ TOP10 Chemically Competent *E. coli* were transformed according to the manufacturer’s instructions using a heat shock method. LB agar (Lennox, Co. Dublin, Ireland) plates containing ampicillin (100 µg/mL) were used to screen for *E. coli* colonies containing each plasmid that were then preserved on cryoprotective beads (TS/71-MX, Protect Multi-purpose, Technical Service Consultants Ltd., Lancashire, UK) and stored at −80 °C. After re-culturing the transformed *E. coli* in ampicillin-containing LB Broth Base (Invitrogen, Thermo Fisher Scientific, Carlsbad, CA, USA), the plasmids carrying the target genes were extracted on a large scale using the GenElute™ HP Plasmid Maxiprep kit (Sigma-Aldrich, St. Louis, MO, USA) and linearised using APA1 restriction enzyme (Promega, Madison, WI, USA) according to the manufacturer’s instructions. The linearised plasmids were purified using the GenElute™ PCR Clean-Up kit (Sigma-Aldrich, St. Louis, MO, USA) and quantified spectrophotometrically. The copy number/μL was determined using the online tool of the URI Genomics & Sequencing Center, which employs the formula mol/g ∗ molecules/mol = molecules/g using Avogadro’s number, 6.022 × 10^23^ molecules/mol (http://cels.uri.edu/gsc/cndna.html, accessed on 14 May 2019). The QPCR reaction volume (20 μL) included 3 μL template DNA, 1 μL of each primer (10 μM), 5 μL of nuclease-free water and 10 μL of Fast SYBR^®^ Green Master Mix (Applied Biosystems, Foster City, CA, USA) for the *Lactobacillus* spp. or GoTaq^®^ qPCR Master Mix (Promega, Madison, WI, USA) for the remaining bacterial groups. All QPCR reactions were carried out in duplicate on the ABI 7500 Fast PCR System (Applied Biosystems, Foster City, CA, USA) with the following cycling conditions; a denaturation step at 95 °C for 10 min, 40 cycles at 95 °C for 15 s and 60 °C for 1 min. PCR assays which exhibited single and specific QPCR products, confirmed by the generation of dissociation curves and visualisation on agarose gel stained with ethidium bromide and with 90–110% efficiency, determined by plotting the threshold cycles (Ct) derived from 5-fold serial dilutions of the plasmid against their arbitrary quantities, were used in this study. Bacterial counts were determined using a standard curve derived from the mean Ct value and the log-transformed gene copy number of the plasmid and expressed as log-transformed gene copy number per gram of faeces (logGCN/g faeces).

### 4.4. Bacterial Strains and Pure Culture Growth Assays

The commensal strains *L. plantarum, L. reuteri* (DSMZ 20016) and *B. thermophilum* and the pathogenic strains *S.* Typhimurium PT12 and enterotoxigenic *E. coli* (ETEC) O149A+ were selected for their beneficial and negative impacts on pig and human health, respectively, and used in the pure culture growth assays as described previously [73]. Briefly, all bacterial strains were revived from cryoprotective beads stored at −20 °C and cultured using standard procedures to obtain 24 h cultures for subsequent use in the pure culture growth assays. *L. plantarum, L. reuteri* and *B. thermophilum* cultures were diluted in 10% de Man, Rogosa and Sharpe broth (MRS, Oxoid Ltd., Hampshire, UK), and *S.* Typhimurium and ETEC cultures were diluted in 10% Tryptone Soya Broth (TSB, Oxoid Ltd., Hampshire, UK) to obtain an inoculum of 10^6^–10^7^ CFU (colony-forming unit)/mL (determined each time). ANWE, ANEE and ANE1–4 were diluted in 10 % MRS and 10% TSB at a working concentration of 4 mg/mL, which were stored at 4 °C and checked regularly for contamination. Further 2-fold dilutions (2–0.25 mg/mL) were performed each time prior to the assay. Next, 100 μL of each extract and dilution in addition to 100 μL inoculum were transferred to duplicate wells of 96-well microtiter plates (CELLSTAR, Greiner Bio-One, Kremsmünster, Austria). Control wells containing 100 μL of 10% medium and 100 μL inoculum were also included. Assay sterility was ensured by including blank wells for each extract and concentration. Plates were agitated gently and incubated aerobically at 37 °C for 18 h, separately from *B. thermophilum,* which was incubated anaerobically. After incubation, bacterial viability and counts were determined by 10-fold serial dilution (10^−1^–10^−8^) and spread plating onto MRS agar (Oxoid Ltd., Hampshire, UK) for *L. plantarum*, *L. reuteri* and *B. thermophilum* and Tryptone Soya Agar (Oxoid Ltd., Hampshire, UK) for ETEC and *S.* Typhimurium. Plates were incubated aerobically at 37 °C for 24 h, with the exception of *B. thermophilum,* which was incubated anaerobically at 37 °C for 48 h. The dilution resulting in 5–50 colonies was selected for the calculation of CFU/mL using the formula CFU/mL = Average colony number ∗ 50 ∗ dilution factor. The bacterial counts were logarithmically transformed (logCFU/mL) for the subsequent statistical analysis. Any zero counts were assigned the arbitrary value of 1.30 logCFU/mL, which was considered the minimum detection limit by spread plating [79]. All experiments were carried out with technical replicates on three independent occasions, hence three biological replicates.

### 4.5. Statistical Analysis

Statistical analysis was performed using Statistical Analysis Software (SAS) 9.4 (SAS Institute, Cary, NC, USA). All data were initially tested for normality using PROC UNIVARIATE procedure.

*Batch fermentation assay:* The counts of the selected bacterial groups in all flasks for each tested compound (*n* = 12, three flasks/each compound concentration) at 10 and 24 h were analysed using the PROC GLM procedure. The statistical model assessed the effect of the compound concentration (0, 1, 2.5 and 5 mg/mL) with the experimental unit being the biological replicate. The counts of each bacterial group at 0 h were used as a covariate. 

*Pure culture growth assay:* The bacterial counts from the pure culture growth assays were analysed using PROC GLM procedure for the presence of linear and quadratic effects of concentration (0, 0.25, 0.5, 1 and 2 mg/mL) for each extract. The biological replicate was the experimental unit. The LSMEANS statement was additionally used to calculate the least-square mean values and the standard error of the means (SEM).

Probability values of <0.05 and <0.10 denote statistical significance and numerical tendency, respectively. Results are presented as least-square mean values ± SEM. 

## 5. Conclusions

In conclusion, the variation in the effects of the whole biomass samples of *A. nodosum*, ANWB-F and ANWB-N, on the counts of total bacteria, *Bifidobacterium* spp. and *Enterobacteriaceae* indicates that the harvest season may influence the presence of the associated bioactive components. Variation in the prebiotic and antibacterial activities of the *A. nodosum* extracts produced using different extraction methods and conditions was also observed, with ANE1 and ANE4 exhibiting the most significant effects. These extracts were produced using different conditions of the HAE methodology with ANE1 resulting in major reductions in *S.* Typhimurium and ETEC counts and ANE4 leading to an increase in *B. thermophilum* counts. Thus, the novel HAE methodology with the E1 and E4 extraction conditions seem to be promising extraction protocols with which to obtain extracts with antibacterial and prebiotic potential, and merit to be further investigated in other seaweed species. The observed bioactivities are most likely attributed to structural alterations of fucoidan due to the different extraction conditions. Purifying these extracts to identify the exact bioactive compound(s) would be beneficial in future studies. As purification processes can be expensive, the antibacterial and prebiotic activity of these extracts should be further confirmed in animal studies as potential nutritional strategies to control intestinal pathogens and promote a more beneficial composition of the gastrointestinal microbiota.

## Figures and Tables

**Table 1 marinedrugs-20-00041-t001:** Proximate composition of whole *A. nodosum* biomass.

Proximate Composition ^1^	Whole Dried *A. nodosum* Biomass	
	ANWB-F	ANWB-N
Dry matter (%)	90.38 ± 0.01	93.94 ± 0.04
Ash (% DW basis)	23.31 ± 0.30	21.87 ± 0.09
Protein (% DW basis)	6.14 ± 0.01	3.52 ± 0.01
Ether extract (% DW basis)	3.33 ± 0.00	2.73 ± 0.03
Total soluble sugars (% DW basis)	13.66 ± 0.08	11.63 ± 0.06
Total glucans (% DW basis)	2.70 ± 0.08	3.30 ± 0.02
Fucoidan (% DW basis)	20.17 ± 0.18	19.50 ± 0.11
Total phenolic content (% DW basis)	0.67 ± 0.01	0.97 ± 0.01

ANWB-F, whole *A. nodosum* biomass sample harvested in February; ANWB-N, whole *A. nodosum* biomass sample harvested in November. ^1^ Results are expressed as mean values ± standard deviation of the mean. The units are expressed as % dry weight (DW) basis.

**Table 2 marinedrugs-20-00041-t002:** Total soluble sugars, laminarin and fucoidan content of *A. nodosum* extracts.

*A. nodosum* Extracts	Extract Composition ^1^		
	Total Soluble Sugars (mg Total Soluble Sugars/100 mg Dry Extract)	Laminarin (mg Total Glucans/100 mg Dry Extract)	Fucoidan (mg Fucoidan/100 mg Dry Extract)
ANWE	8.95 ± 0.72	ND	2.44 ± 0.09
ANEE	6.64 ± 0.47	ND	0.80 ± 0.05
ANE1	ND	1.31 ± 0.06	26.75 ± 0.12
ANE2	ND	1.57 ± 0.10	29.97 ± 0.60
ANE3	ND	0.99 ± 0.12	28.53 ± 0.48
ANE4	ND	1.26 ± 0.03	27.44 ± 0.08

ANWE, *A. nodosum* water extract; ANEE, *A. nodosum* ethanol extract; ANE1–4, *A. nodosum* extract 1–4; ND, not determined. ^1^ Results are expressed as mean values ± standard deviation of the mean.

**Table 3 marinedrugs-20-00041-t003:** Effects of FOS, ANWB-F and ANWB-N on the counts of selected bacterial groups at 10 and 24 h in the batch fermentation assay (Least-square mean values with their standard errors).

Compound	Time Point	Bacterial Group (logGCN/g Faeces)	Compound Concentration (mg/mL)				SEM	*p*-Value
			0	1	2.5	5		
FOS	10 h	Total bacteria	8.69 ^a^	9.13 ^ab^	9.54 ^b^	9.49 ^b^	0.149	0.017
		*Lactobacillus* spp.	8.64	8.71	8.74	8.78	0.080	0.703
		*Bifidobacterium* spp.	6.39	6.48	6.72	6.70	0.080	0.053
		*Enterobacteriaceae*	7.73	7.79	8.08	7.67	0.145	0.284

	24 h	Total bacteria	9.63	9.78	9.77	9.95	0.112	0.321
		*Lactobacillus* spp.	8.69	8.76	8.69	8.77	0.106	0.923
		*Bifidobacterium* spp.	6.68	6.69	6.68	6.79	0.048	0.323
		*Enterobacteriaceae*	7.75	7.64	7.60	7.30	0.260	0.659
ANWB-F	10 h	Total bacteria	8.92	9.31	9.36	9.41	0.206	0.374
		*Lactobacillus* spp.	8.60	8.79	8.71	8.68	0.094	0.577
		*Bifidobacterium* spp.	6.42	3.70	1.80	U/D	1.357	0.057
		*Enterobacteriaceae*	7.58	7.75	7.70	7.54	0.098	0.447

	24 h	Total bacteria	9.48	9.82	9.84	9.97	0.162	0.268
		*Lactobacillus* spp.	8.68	8.75	8.76	8.76	0.104	0.938
		*Bifidobacterium* spp.	6.87 ^b^	5.92 ^b^	1.57 ^a^	U/D ^a^	0.718	0.001
		*Enterobacteriaceae*	7.99	8.10	7.83	7.68	0.150	0.297
ANWB-N	10 h	Total bacteria	8.12	9.00	8.99	9.13	0.228	0.055
		*Lactobacillus* spp.	8.62	8.75	8.70	8.52	0.080	0.290
		*Bifidobacterium* spp.	6.39 ^b^	6.46 ^b^	1.64 ^a^	U/D ^a^	0.826	0.002
		*Enterobacteriaceae*	6.99	7.09	7.08	6.90	0.108	0.603

	24 h	Total bacteria	8.95 ^a^	9.58 ^b^	9.69 ^b^	9.67 ^b^	0.127	0.013
		*Lactobacillus* spp.	8.66 ^ab^	8.82 ^b^	8.82 ^b^	8.60 ^a^	0.047	0.030
		*Bifidobacterium* spp.	6.79 ^c^	5.03 ^b^	U/D ^a^	U/D ^a^	0.235	<0.001
		*Enterobacteriaceae*	7.26 ^b^	7.58 ^c^	7.51 ^bc^	6.83 ^a^	0.077	0.001

FOS, fructooligosaccharides; ANWB-F, whole *A. nodosum* biomass sample harvested in February; ANWB-N, whole *A. nodosum* biomass sample harvested in November; logGCN/g faeces, log transformed gene copy number per gram faeces; U/D, undetectable. ^a,b,c^ Mean values within a row with different superscript letter were significantly different (*p* < 0.05).

**Table 4 marinedrugs-20-00041-t004:** Bacterial counts following exposure to increasing concentrations of the *A. nodosum* extracts in the pure culture growth assays (Least-square mean values with their standard errors).

*A. nodosum* Extract	Bacterial Strain	Final Bacterial Concentration (logCFU/mL)					SEM	Linear Effect *p*-Value	Quadratic Effect *p*-Value
		0 mg/mL ^1^	0.25 mg/mL ^1^	0.5 mg/mL ^1^	1 mg/mL ^1^	2 mg/mL ^1^			
ANWE	*L. plantarum*	7.55	7.17	6.89	6.96	7.10	0.101	<0.001	**0.001**
	*L. reuteri*	7.06	5.70	5.17	5.66	1.68	0.224	0.422	**0.032**
	*B. thermophilum*	6.67	6.82	6.97	7.24	7.26	0.128	0.002	**0.050**
	*S.* Typhimurium	9.09	9.15	9.16	9.20	9.36	0.066	**0.001**	0.950

ANEE	*L. plantarum*	7.22	7.75	7.90	7.81	7.97	0.123	**0.005**	0.132
	*L. reuteri*	7.12	6.07	5.92	4.78	1.43	0.172	**<0.001**	0.066
	*B. thermophilum*	6.80	6.71	5.92	4.51	2.80	0.368	**<0.001**	0.645
	*S.* Typhimurium	8.73	8.92	8.96	8.94	8.92	0.071	0.382	0.193

ANE1	*L. plantarum*	7.89	7.91	7.95	7.91	7.96	0.072	0.373	0.813
	*L. reuteri*	7.59	7.42	7.71	7.72	7.78	0.084	**0.036**	0.630
	*B. thermophilum*	6.57	6.55	6.46	6.24	5.63	0.150	**<0.001**	0.364
	*ETEC*	8.26	8.36	8.23	8.14	3.36	0.105	<0.001	**<0.001**
	*S.* Typhimurium	8.88	8.65	8.98	8.76	3.41	0.093	<0.001	**<0.001**

ANE2	*L. plantarum*	7.70	7.83	7.83	8.01	7.90	0.083	0.125	0.085
	*L. reuteri*	7.17	7.08	7.21	7.15	7.11	0.081	0.992	0.583
	*B. thermophilum*	6.69	6.56	6.69	6.26	5.62	0.052	0.423	**0.016**
	*ETEC*	8.42	8.96	8.87	8.56	7.44	0.143	0.033	**0.001**
	*S.* Typhimurium	8.94	9.06	9.08	8.96	8.79	0.067	**0.011**	0.117

ANE3	*L. plantarum*	7.87	7.88	7.90	7.93	7.92	0.058	0.488	0.690
	*L. reuteri*	7.31	7.37	7.34	7.24	7.28	0.099	0.217	0.531
	*B. thermophilum*	6.69	6.54	6.55	6.47	6.53	0.168	0.768	0.602
	*ETEC*	8.40	8.45	8.58	8.56	8.63	0.070	0.235	0.806
	*S.* Typhimurium	8.98	8.97	8.87	9.06	8.90	0.039	0.549	0.345

ANE4	*L. plantarum*	7.95	7.98	8.03	8.16	8.08	0.052	0.225	0.179
	*L. reuteri*	7.50	7.65	7.55	7.61	7.70	0.064	**0.043**	0.921
	*B. thermophilum*	6.48	7.01	7.13	7.35	7.37	0.112	<0.001	**0.004**
	*ETEC*	8.66	8.56	8.58	8.75	8.68	0.098	0.300	0.672
	*S.* Typhimurium	9.11	9.19	9.09	9.16	9.01	0.052	0.168	0.185

ANWE, *A. nodosum* water extract; ANEE, *A. nodosum* ethanol extract; ANE1–4, *A. nodosum* extract 1–4; CFU, colony-forming unit. ^1^ The concentrations tested for each *A. nodosum* extract in the pure culture growth assays.

**Table 5 marinedrugs-20-00041-t005:** Preparation of whole *A. nodosum* biomass samples, with extraction methods and conditions employed to obtain the different *A. nodosum* extracts.

*A. nodosum* Sample	Extraction Method ^1^	Solvent ^1^	Conditions ^1^	Optimised for Targeted Bioactives
ANWB-F ANWB-N	N/A	N/A	Oven-dried at 50 °C for 9 days and milled to 1 mm particle size	N/A
*Conventional extraction methods*				
ANWE	solvent extraction	water	Room temperature (20 °C) 24 h 20 mL solvent/g seaweed Stirring at 170 rpm	crude
ANEE	solvent extraction	80% ethanol 20% water	Room temperature (20 °C) 24 h 10 mL solvent/g seaweed Stirring at 170 rpm	polyphenols
*Hydrothermal-assisted extraction methodology*				
ANE1	HAE	0.1 M HCl	120 °C 62.1 min 30 mL solvent/g seaweed	Fucoidan
ANE2	HAE	0.1 M HCl	99.3 °C 30 min 21.3 mL solvent/g seaweed	Laminarin
ANE3	HAE	0.1 M HCl	120 °C 76.06 min 10 mL solvent/g seaweed	Antioxidant activity
ANE4	HAE	0.1 M HCl	120 °C 80.9 min 12.02 mL solvent/g seaweed	For laminarin, fucoidan and antioxidant activity

ANWB-F, whole *A. nodosum* biomass sample harvested in February; ANWB-N, whole *A. nodosum* biomass sample harvested in November; ANWE, *A. nodosum* water extract; ANEE, *A. nodosum* ethanol extract; ANE1–4, *A. nodosum* extract 1–4; HAE, hydrothermal-assisted extraction; N/A, not applicable. ^1^ Extraction methods and conditions as described by Tierney et al. [69] and Garcia-Vaquero et al. [26].

**Table 6 marinedrugs-20-00041-t006:** List of forward and reverse primers used for the bacterial quantification by QPCR. Bp: base pairs; Tm: melting temperature.

Target Bacterial Group	Reverse Primer (5′-3′) Forward Primer (5′-3′)	Amplicon Length (bp)	Tm (°C)	References
Total bacteria	F: GTGCCAGCMGCCGCGGTAA R: GACTACCAGGGTATCTAAT	291	64.2 52.4	[75]
*Lactobacillus* spp.	F: AGCAGTAGGGAATCTTCCA R: CACCGCTACACATGGAG	341	54.5 55.2	[76]
*Bifidobacterium* spp.	F: GCGTGCTTAACACATGCAAGTC R: CACCCGTTTCCAGGAGCTATT	125	60.3 59.8	[77]
*Enterobacteriaceae*	F: ATGTTACAACCAAAGCGTACA R: TTACCYTGACGCTTAACTGC	185	54.0 56.3	[78]

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
