# Peer review of "Evaluation of the Antibacterial and Prebiotic Potential of Ascophyllum nodosum and Its Extracts Using Selected Bacterial Members of the Pig Gastrointestinal Microbiota"

_marinedrugs, 2021, doi:10.3390/md20010041_

Round 1
Reviewer 1 Report
In this manuscript were evaluated the antibacterial and prebiotic effects of
Ascophyllum nodosum extracts using selected bacteria of the pig gastrointestinal microbiota. It is well written and organized showing the influence of the collection time and the influence of the extraction methodology. However, there are some aspects that should be clarified, in particular, the authors analyzed the effects of a collection of the extracts in February and November but did not say if performed statistical analysis. In my opinion, this statistical analysis should be performed. Regarding the statistical analysis, it was not clear which is the p-value linear and quadratic effects.
There are minor points that should be corrected:
- Page 7 should be table 4 and not table 3;
- In the table 4 it were not evident that the concentrations refers to the extract;
- Please put S. Typhimurium in italic.
Author Response
Point 1. However, there are some aspects that should be clarified, in particular, the authors analyzed the effects of a collection of the extracts in February and November but did not say if performed statistical analysis. In my opinion, this statistical analysis should be performed.
Response 1. Thank you for your suggestion. We assume that you refer to the proximate composition of the whole biomass samples of A. nodosum. This data has previously been analysed and published as referenced in section 2.1 of the results in this manuscript. Therefore, no further statistical analysis on the composition of the A. nodosum samples was performed and the comparisons presented in this section of the manuscript were removed, as this information is thoroughly presented in the study by Garcia-Vaquero et al, 2021.
Point 2. Regarding the statistical analysis, it was not clear which is the p-value linear and quadratic effects.
Response 2. Thank you for this comment. We assume that there was an issue with the presentation of the data in Table 4. Therefore, the p-value of the strongest effect (quadratic>linear) for the statistically significant differences in bacterial counts is in bold text to improve the presentation of the results.
Point 3. Page 7 should be table 4 and not table 3;
Response 3. Corrected as requested.
Point 4. In the table 4 it were not evident that the concentrations refers to the extract;
Response 4. A footnote was added in Table 4 to clarify that each A. nodosum extract was evaluated at the concentrations presented in the pure culture growth assays.
Point 5. Please put S. Typhimurium in italic.
Response 5. Thank you for your suggestion. According to the recommended nomenclature for the Salmonella genus, the serotypes of Salmonella enterica subsp. enterica should not be italicised but only start with a capital letter (Brenner, F.W., Villar, R.G., Angulo, F.J., Tauxe, R. and Swaminathan, B., 2000. Salmonella nomenclature. Journal of clinical microbiology, 38(7), pp.2465-2467).
Reviewer 2 Report
Venardou and co-workers presente the manuscript entitled “Evaluation of the Antibacterial and Prebiotic Potential of Ascophyllum nodosum and its Extracts Using Selected Bacterial Members of the Pig Gastrointestinal Microbiota”, where they explore the whole biomass and different extracts of the seaweed A. nodosum in the modulation of pigs intestinal microbiota, aiming to explore its potential as prebiotics.
This is an interesting study, however, some issues should be addressed:
- The authors determined the proximate composition of whole A. nodosum biomass, from February and November. The statistical analysis must be performed for results comparison, and the p values must be includes in the table and along the text. For instance, in lines 90-91, authors say that “protein concentration was higher in February compared to November (6.14% vs 3.52%)”, the statistical analysis must be performed to add significance to the study.
- Table 2: replace “freeze-dried extract” for “dry extract”. Also include the statistical analysis for results comparison within each column.
- Please correct “S. Typhimurium” to “S. typhimurium”.
- “P” value should be written lower case and italics (p).
- ANWE and ANEE extracts reduce the counts of L. reuteri. ANE1 reduces the final bacterial concentration of S. typhimurium for the highest concentration tested (2 mg/mL). What do the authors expect, based on the extracts composition, that may be responsible for this behavior?
- Lines 213-215: According to Table 3, there is no information on the increase of bifidobacterium with FOS (p>0.05), and the increase in total bacteria only occurs at 10h and for the lowest concentrations. Please discuss.
- Lines 235-239: Authors cannot make this assumption based on the results of the present survey; a characterization of fucoidan in samples from February and November should be performed. A discussion based on the differences observed by the authors in samples composition (Table 1) should be performed.
- Lines 280-283: The antibacterial and bifidogenic activities of ANE1 and ANE4 are attributed to Fucoidan. Based on Table 2, ANE2 and ANE3 have similar Fucoidan amounts, but different activity. Please clearly explain the differences expected in Fucoidan structure according to each extraction condition, and how their affect the bioactivity.
- Please review tables numbering, there are 2 “Table 3”.
Author Response
Point 1. The authors determined the proximate composition of whole A. nodosum biomass, from February and November. The statistical analysis must be performed for results comparison, and the p values must be includes in the table and along the text. For instance, in lines 90-91, authors say that “protein concentration was higher in February compared to November (6.14% vs 3.52%)”, the statistical analysis must be performed to add significance to the study.
Response 1. Thank you for your suggestion. The data on the proximate composition of the whole biomass samples of A. nodosum has previously been analysed and published as referenced in section 2.1 of the results in this manuscript. Therefore, no further statistical analysis on the composition of the A. nodosum samples was performed and the comparisons presented in this section of the manuscript were removed, as this information is thoroughly presented in the study by Garcia-Vaquero et al, 2021.
Point 2. Table 2: replace “freeze-dried extract” for “dry extract”. Also include the statistical analysis for results comparison within each column.
Response 2. The word “freeze-dried” was replaced by the word “dry” as requested. Table 2 provides information on the content of the A. nodosum extracts for the selected sugars. Therefore, no statistical analysis was performed and the relevant text in section 2.1 of the manuscript was edited accordingly.
Point 3. Please correct “S. Typhimurium” to “S. typhimurium”.
Response 3. Thank you for your suggestion. According to the recommended nomenclature for the Salmonella genus, the serotypes of Salmonella enterica subsp. enterica should not be italicised but only start with a capital letter (Brenner, F.W., Villar, R.G., Angulo, F.J., Tauxe, R. and Swaminathan, B., 2000. Salmonella nomenclature. Journal of clinical microbiology, 38(7), pp.2465-2467).
Point 4. “P” value should be written lower case and italics (p).
Response 4. Corrected as requested.
Point 5. ANWE and ANEE extracts reduce the counts of L. reuteri. ANE1 reduces the final bacterial concentration of S. typhimurium for the highest concentration tested (2 mg/mL). What do the authors expect, based on the extracts composition, that may be responsible for this behavior?
Response 5. Due to the poor performance of ANWE and ANEE in the pure culture growth assays, the composition of these extracts was not further explored beyond the total soluble sugar and fucoidan content which were present in low concentrations. Therefore, we cannot make any assumptions for which component is responsible for the bioactivities of these two extracts.
In the case of ANE1 we assumed that the observed antibacterial activity was due to fucoidan based on its relatively high concentration in this extract. However, we did not perform a full compositional characterisation of ANE1 to confirm this assumption, as the purpose of the current study was to screen several A. nodosum extracts to identify the ones with promising antibacterial and/or prebiotic activities.
Point 6. Lines 213-215: According to Table 3, there is no information on the increase of bifidobacterium with FOS (p>0.05), and the increase in total bacteria only occurs at 10h and for the lowest concentrations. Please discuss.
Response 6. The p-value of 0.053 (tendency) presented in Table 3 was considered indicative of an increase in the counts of bifidobacteria at 10h at 2.5 and 5 mg/ml FOS as described in section 2.2 of the results in this manuscript. Regarding the increase in total bacterial counts at 10h at 2.5 and 5 mg/ml FOS, but not at 24h, this is probably due to the accumulation of toxic bacterial metabolites and the depletion of nutrients as a result of the bacterial growth which characterise batch fermentation assays.
Point 7. Lines 235-239: Authors cannot make this assumption based on the results of the present survey; a characterization of fucoidan in samples from February and November should be performed. A discussion based on the differences observed by the authors in samples composition (Table 1) should be performed.
Response 7. That is correct. As the purpose of this initial screening study was to explore the antibacterial and prebiotic potential of A. nodosum, whole biomass samples and extracts, no characterisation of the fucoidan was performed. The relevant part of the discussion (lines 222-226) has been edited accordingly to be in line with the analysis performed and the results presented in this study.
Point 8. Lines 280-283: The antibacterial and bifidogenic activities of ANE1 and ANE4 are attributed to Fucoidan. Based on Table 2, ANE2 and ANE3 have similar Fucoidan amounts, but different activity. Please clearly explain the differences expected in Fucoidan structure according to each extraction condition, and how their affect the bioactivity.
Response 8. Thank you for this comment. In the discussion of our manuscript, we assumed that the antibacterial and prebiotic activities of the ANE1 and ANE4, respectively, were associated with structural changes in fucoidan due to the different extraction conditions, as the concentration of fucoidan in ANE1-4 was at similar levels. However, we cannot speculate on the expected level of depolymerisation or the expected changes in the chemical structure of fucoidan caused by the extraction conditions used for each extract, mainly temperature and solvent. As this was an initial screening study to identify extracts with antibacterial and/or prebiotic potential, future work should include further compositional characterisation of these extracts and chemical and structural analysis of fucoidan that would provide a clear answer on this. The relevant part of the discussion (lines 267-271) was edited accordingly to be in line with the analysis performed and results presented in this study.
Point 9. Please review tables numbering, there are 2 “Table 3”.
Response 9. Corrected as requested.
Round 2
Reviewer 2 Report
The authors have answered the questions accordingly and the manuscript is suitable for publication.